# MG-LLaVA: Towards Multi-Granularity Visual Instruction Tuning

## Abstract

Multi-modal large language models (MLLMs) have made significant strides in various visual understanding tasks. However, the majority of these models are constrained to process low-resolution images, which limits their effectiveness in perception tasks that necessitate detailed visual information. In our study, we present MG-LLaVA, an innovative MLLM that enhances the model's visual processing capabilities by incorporating a multi-granularity vision flow, which includes low-resolution, high-resolution, and object-centric features. We propose the integration of an additional high-resolution visual encoder to capture fine-grained details, which are then fused with base visual features through a Conv-Gate fusion network. To further refine the model's object recognition abilities, we incorporate object-level features derived from bounding boxes identified by offline detectors. Being trained solely on publicly available multimodal data through instruction tuning, MG-LLaVA demonstrates exceptional perception skills. We instantiate MG-LLaVA with a wide variety of language encoders, ranging from 3.8B to 34B, to evaluate the model's performance comprehensively. Extensive evaluations across multiple benchmarks demonstrate that MG-LLaVA outperforms existing MLLMs of comparable parameter sizes, showcasing its remarkable efficacy.

## 1 Introduction

Recent works on Multimodal Large Language Models (MLLMs) (Zhu et al., 2023; Ye et al., 2023; Liu et al., 2024b; Zhang et al., 2023b; Wei et al., 2023; Xu et al., 2023) have achieved rapid development in vision language understanding, visual reasoning, visual interaction, and localization. Most MLLMs adopt pre-trained Large Language Models (LLMs) as the base architecture to process concatenated visual and language embeddings. As one representative work, LLaVA (Liu et al., 2024b) adopts low-resolution ($224^2$, $336^2$, *etc.*) images as inputs and aligns visual embeddings with the text modality via an MLP projector and then performs instruction tuning. The architecture of LLaVA has been widely adopted by subsequent works (Xu et al., 2024; Li et al., 2024c; Maaz et al., 2023; Lin et al., 2023a), and has been applied to various vision tasks, including detection, segmentation, and video understanding.

Real-world images exhibit a wide range of resolutions, scales, and aspect ratios, posing significant challenges for MLLMs with low-resolution inputs in robustly processing them. To tackle this problem, recent works (Liu et al., 2024a; Lin et al., 2023b; Li et al., 2024c; Zong et al., 2024; Luo et al., 2024; Xu et al., 2024; Dong et al., 2024) have proposed various strategies to augment the capabilities of visual encoders in MLLMs, including training on diverse datasets, utilizing high-resolution image inputs, and employing dynamic aspect ratios. Most of these approaches involve the integration of additional visual tokens through various techniques. Despite these advancements, two critical issues persist: (1) Although object-level features are crucial in nearly all visual understanding tasks, they are currently absent in existing vision encoders; (2) None of the existing MLLMs have integrated multi-granularity features, a classic concept in computer vision, into their frameworks. However, as a human vision system, multi-granularity inputs are common in various cases since even on the same object, the scale variance problems pose challenges (Ren et al., 2015; Ghiasi et al., 2019) in the current perception system.

Motivated by the aforementioned analysis, we introduce MG-LLaVA, a novel MLLM designed to effectively process multi-granularity visual inputs, including object-level, origin images, and

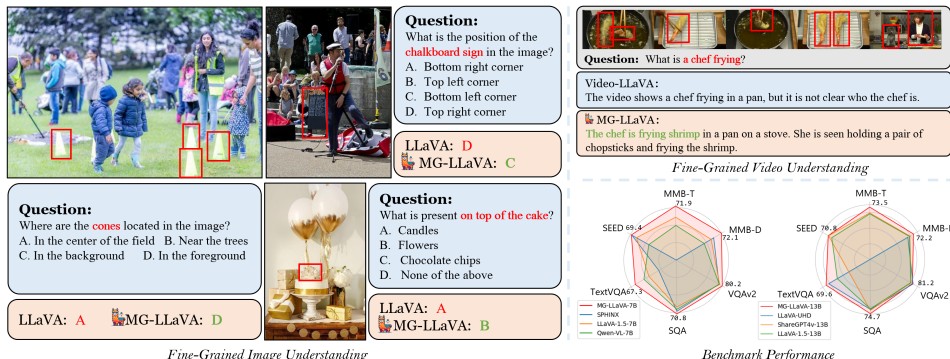

Figure 1: MG-LLaVA demonstrates notable performance across various vision-language tasks, particularly on tasks involving object recognition.

high-resolution inputs. Our framework builds upon LLaVA (Liu et al., 2024b) and is specifically tailored to incorporate and manage multi-granularity inputs. For object-level inputs, we employ a pre-trained open-vocabulary detector to identify object bounding boxes and execute region features to acquire region visual tokens. In particular, we explore two methods for object feature integration: explicit integration via box feature fusion and implicit integration via object proposal feature. We find that the former works well and it can even scale up with more data. In contrast to close-set detectors, open-vocabulary detectors offer enhanced generalizability and robustness across diverse scenes. To handle fine-grained visual inputs, we utilize a convolution-based backbone Schuhmann et al. (2022) to extract richer visual features. Subsequently, we propose a straightforward yet effective fusion strategy to integrate these inputs into the original visual tokens in LLaVA. Specifically, we initially merge the fine-grained visual tokens with the original visual tokens using a simple Conv-Gate convolution. Then, we append the object-level tokens to the fused tokens. Fig. 2 illustrate the difference between MG-LLaVA and existing MLLMs. Experimental results quantitatively validate the efficacy of the design of MG-LLaVA.

We perform extensive experiments with MG-LLaVA integrated with various language encoders, ranging from 3.8B to 34B, to substantiate the effectiveness of MG-LLaVA. Our evaluation encompasses 13 popular multimodal benchmarks for both image and video. Additionally, we present a comprehensive set of ablation studies that illustrate the impact of different components in MG-LLaVA. Benefiting from multi-granularity visual features, MG-LLaVA demonstrates a significantly enhanced capability in perception and visual comprehension, outperforming established counterparts and notably surpassing GPT-4V (OpenAI, 2023) and GeminiPro-V (Team et al., 2023) on various multimodal benchmarks, including MMBench (Liu et al., 2023c) and SEEDBench (Li et al., 2023a).

The contribution of this work can be summarized as follows:

• We introduce MG-LLaVA, an advanced multi-modal model adept at processing visual inputs of multiple granularities, including object-level features, original-resolution images, and high-resolution data. This advancement significantly enhances the capabilities of MLLMs in visual perception and understanding.
• We propose the Multi-Granularity Vision Flow, a straightforward yet effective module designed to integrate features across various granularities, thereby significantly improving the performance of our model. The effectiveness of our approach is substantiated through empirical experiments.
• By employing a range of language models scaling from 3.8B to 34B, our model exhibits clear scalability and a marked proficiency in visual comprehension, outperforming established counterparts and notably surpassing GPT-4V and GeminiPro-V on MMBench and SEEDBench.

## 2 RELATED WORK

**Large Language Models.** In recent years, private large language models (LLMs) like GPT-4 (OpenAI, 2023) and Llama (Touvron et al., 2023) have gained remarkable performance. Concurrently,

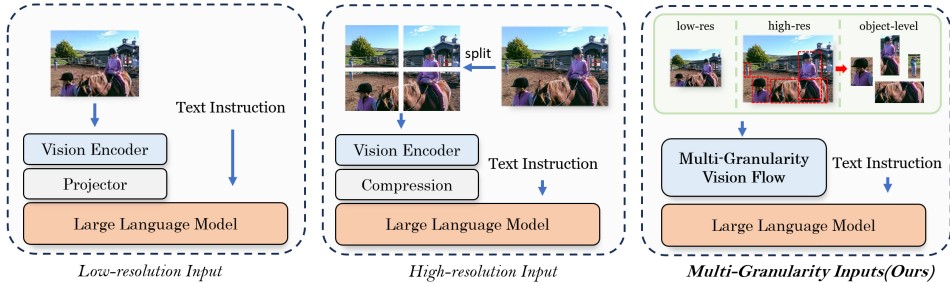

Figure 2: **Comparing Different MLLM Paradigms**. MG-LLaVA effectively perceives multi-granularity visual inputs that include object-level, low, and high-resolution inputs, thereby achieving advanced multi-modal understanding.

a multitude of open-source research (Chiang et al., 2023; Yang et al., 2023; Bai et al., 2023; Team, 2023) has embarked on the exploration of LLMs. LLM shows strong performance in various NLP tasks. However, pure LLMs cannot handle image and video inputs. Our work focuses on designing new multimodal large language models, which jointly take visual and language tokens as inputs. In this work, we engaged a range of LLMs (Chiang et al., 2023; Abdin et al., 2024; AI@Meta, 2024; Young et al., 2024) scaling from 3.8B to 34B. The observed performance across these models has proved the effectiveness of our design.

**Multimodal Large Language Models.** Multi-modal Large Language Models (MLLMs) (Zhu et al., 2023; Ye et al., 2023; Chen et al., 2023c; Dai et al., 2024; Bai et al., 2023; Liu et al., 2023a; Li et al., 2023c; Lin et al., 2023a; Zhang et al., 2024; Huang et al., 2024; Wu et al., 2024) have recently showcased the potential to endow LLMs with visual conversational abilities. Among these models, LLaVA (Liu et al., 2023a) typically built a simple architecture that utilizes a vision-language cross-modal adapter to bridge the gap between vision and language tokens. Some research (Li et al., 2023d; Zhang et al., 2023c; Liu et al., 2024a) tried to increase performance by utilizing high-resolution inputs. LLaVA-UHD (Xu et al., 2024) cost-effectively increased input resolution by dividing high-resolution images into smaller slices. Subsequently, LLaVA-HR (Luo et al., 2024) and Mini-Gemini (Li et al., 2024c), endeavor to incorporate an additional visual encoder to enhance high-resolution details without increasing the count of visual tokens. However, these works consistently overlook the impact of fine-grained object-level features, which compromises their potential for enhanced perception. In comparison, MG-LLaVA explores the potential of multi-granularity input by simultaneously leveraging high-resolution inputs, low-resolution inputs, and object-level inputs. By flexibly integrating visual tokens of multiple granularity, MG-LLaVA achieves superior performance on several benchmarks with a marginal increase in cost.

**Multi-Granularity Modeling in Vision.** Inputs of multiple granularity have been incorporated into various downstream vision tasks. In object detection and segmentation, the efficacy of multi-level features has been well-established in detecting objects of different scales (Zhao et al., 2019a; Qian et al., 2021; Liu et al., 2023b; Wan et al., 2019; Li et al., 2024b; Yuan et al., 2024; Zhou et al., 2023). For panoptic segmentation, some methods (de Geus et al., 2019; Kirillov et al., 2019; Li et al., 2019; Xu et al., 2022; Ramanathan et al., 2023; Qi et al., 2024) applied a multi-granularity network to train instance, semantic, and part segmentation in parallel, and some studies (Michieli et al., 2020; Zhao et al., 2019b; de Geus et al., 2021; Li et al., 2022; 2024a) have indicated that training on various levels of abstraction can improve the performance of the segmentation network. For example, SAM (Kirillov et al., 2023) presents a multi-granularity mask prediction method for handling various level masks, such as things, background stuff, and parts. Motivated by the above works, we aim to capture input from various levels of perception into MLLM. In particular, we construct our model by developing multiple visual branches for different granularity, thereby augmenting its perceptual capabilities.

## 3 METHOD

In this work, we propose MG-LLaVA, effectively harnesses both the high-resolution and object-level features for improving MLLMs. The architecture of MG-LLaVA is illustrated in Fig. 3a. The model

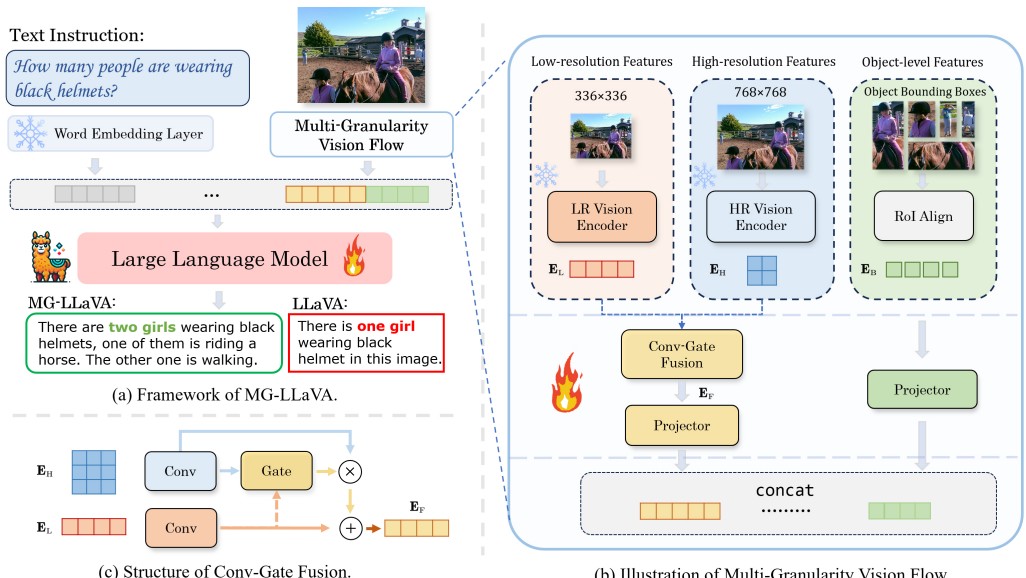

Figure 3: The illustration of MG-LLaVA. **Top left**: The overall framework of MG-LLaVA, which includes the Multi-Granularity Vision Flow module and a LLM. **Right**: Illustration of Multi-Granularity Vision Flow, which aims to extract multiple visual features and integrate disparate features to ensure seamless interaction. **Botttom left**: Structure of Conv-Gate Fusion module.

comprises two key components: (1) Multi-Granularity Vision Flow framework for extracting visual features with different resolutions and granularities while effectively integrating disparate features to ensure seamless interaction. (2) A large language model dedicated to generating coherent and contextually relevant responses.

## 3.1 PRELIMINARY

As one of the most extensively adopted multi-modal LLM architectures, LLaVA consists of a vision encoder $f_{\text{V}}$, an MLP projector $f_p$, and a language model $f_{\text{L}}$. Given a visual input $V$ and a textual input $T$, LLaVA computes the vision and language embeddings as per Eq. (1), where $f_{\text{T}}$ represents the input embedding layer of $f_{\text{L}}$. The resulting embeddings, $\mathbf{E}_{\text{T}}$ and $\mathbf{E}_{\text{V}}$, are then concatenated into a single token sequence, serving as the input to the LLM. LLaVA utilizes Eq. (2) to calculate the probability of the target answer $\mathbf{X}_{\text{A}}$, where $\theta$ represents the trainable parameters and $L$ is the length of $\mathbf{X}_{\text{A}}$. The model is trained on visual instruction tuning data to maximize $p\left(\mathbf{X}_{\text{A}} \mid V, T\right)$.

$$\mathbf{E}_{\text{T}} = f_{\text{T}}\left(T\right), \mathbf{E}_{\text{V}} = f_p\left(f_{\text{V}}\left(V\right)\right) \tag{1}$$

$$p\left(\mathbf{X}_{\text{A}} \mid V, T\right) = \prod_{i=1}^{L} p_\theta\left(\mathbf{X}_{\text{A}}^{[i]} \mid Concat(\mathbf{E}_{\text{V}}, \mathbf{E}_{\text{T}}^{[1:i-1]}), \mathbf{X}_{\text{A}}^{[i-1]}\right) \tag{2}$$

Despite the promising results, LLaVA still restrains itself in processing images at a low resolution ($224^2$, $336^2$, *etc.*), This significantly hinders the model's ability, particularly in recognizing small objects. Scaling to high resolution without adapting the vision encoder directly would dramatically increase the number of visual tokens, rendering the approach ineffective. Furthermore, the visual input can also be complex and contain numerous objects within an image or video, which poses challenges for MLLMs in identifying some critical objects. Empirically, incorporating object-level features can significantly enhance the model's perceptual abilities. Therefore, we introduce MG-LLaVA, which effectively harnesses both the high-resolution and object-level features for the improvement of MLLMs.

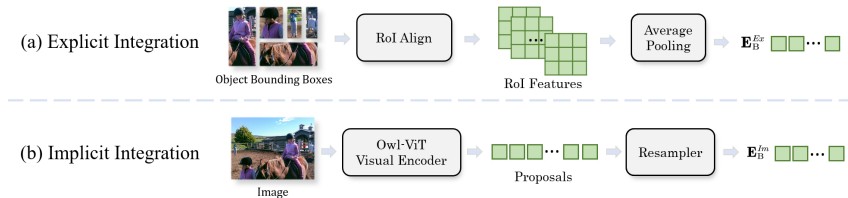

Figure 4: Comparison of explicit and implicit integration of object-level features.

## 3.2 MULTI-GRANULARITY VISION FLOW

**Hybrid Vision Encoders**   As depicted in Fig. 3b, MG-LLaVA initially processes images at two different resolutions: low-resolution $V_L$ and high-resolution $V_H$. In the low-resolution branch, we follow the LLaVA-1.5 (Liu et al., 2023a) to utilize a CLIP-pretrained ViT (Radford et al., 2021) denoted as $f_V^L$ to derive low-resolution features $\mathbf{E}_L \in \mathbb{R}^{N \times C}$. The ViT feature $\mathbf{E}_L$ benefits from an expanded receptive field, capturing a more comprehensive view of global information. In the high-resolution branch, we employ a CLIP-pretrained ConvNeXt (Schuhmann et al., 2022) denoted by $f_V^H$ to obtain high-resolution features $\mathbf{E}_H \in \mathbb{R}^{h \times w \times C}$. $f_V^H$ effectively extracts detailed features from high-resolution images, offering detailed local insights. $f_V^L$ and $f_V^H$ downsample the input resolution with strides of 14 and 32, respectively. We therefore adjust $V_L$ and $V_H$ to ensure that the number of tokens in $\mathbf{E}_L$ and $\mathbf{E}_H$ remains the same ($N = h \times w$).

**Conv-Gate Fusion**   Combining both low and high-resolution features as inputs results in a doubling of the visual tokens to be processed, which is computationally ineffective. Moreover, the distinct architectures of ViT and ConvNeXt lead to a discrepancy between $\mathbf{E}_L$ and $\mathbf{E}_H$, requiring a careful fusion process. Inspired from (Luo et al., 2024), we implement a lightweight Conv-Gate fusion network that facilitates feature aggregation while maintaining a single resolution's token count, as shown in Fig. 3c. We first employ 1D convolutions to align the channel widths of heterogeneous features and subsequently use a gating layer to modulate the semantic information across low and high resolutions, as described in Eq. (3). The fusion module is applied to the output of both vision encoders, resulting in only a marginal increase in computational cost.

$$\mathbf{E}_F = \mathbf{E}_L + G(Conv(\mathbf{E}_L), Conv(\mathbf{E}_H)) \times \mathbf{E}_H \tag{3}$$

**Integration of Object-level Features**   We investigate the integration of object-level features through both explicit and implicit methodologies.

(1) Explicit integration. We first employ an offline detector to delineate the bounding boxes of objects within the image. Given the set of $k$ object bounding boxes derived from the image, denoted as $B = \{b_1, b_2, \cdots, b_k\}$, we employ the Region of Interest (RoI) Align to extract object-level features from the vision features of the high-resolution encoder $f_V^H$. Specifically, we upsample and concatenate features from different convolutional stages to a scale of 1/4 the input size, resulting in a multi-scale feature representation $f_V^{H\prime}$, which provides a fine-grained perspective. The object-level features are then aligned from $f_V^{H\prime}$. To maintain computational efficiency, we apply average pooling to each object feature and subsequently concatenate them into a sequence $\mathbf{E}_B^{Ex} \in \mathbb{R}^{k \times C}$, as detailed in Eq. (4). The progress is illustrated in Fig. 4a.

$$\mathbf{E}_B^{Ex} = Concat(Avg(RoIAlign(f_V^{H\prime}, B))) \tag{4}$$

(2) Implicit integration. We propose the implicit integration of object-level features by incorporating proposal information. Owl-ViT-v2 (Minderer et al., 2024) is a robust detector that utilizes its visual encoder to generate proposals of the objects within the input image. Given an image $I$, the output of Owl-encoder $f_O$ is represented as $P_O \in \mathbb{R}^{L \times D}$, where $L$ denotes the number of proposals and $D$ denotes the dimension of the output. Each proposal can be interpreted as a potential object within the image, encompassing information regarding its position and category. Given the substantial number of proposals(in the thousands), we utilize a resampler module (Alayrac et al., 2022), denoted as $S$,

to extract the information from the output proposals, represented as $\mathbf{E}_{\mathrm{B}}^{Im} \in \mathbb{R}^{L' \times D}$. The number of output queries $L'$ generated by the resampler is significantly fewer than the output proposals $L$ produced by the Owl-encoder. The entire progress is depicted in Fig. 4b, as described in Eq. (5).

$$\mathbf{E}_{\mathrm{B}}^{Im} = S(f_O(I)) \tag{5}$$

In our experiments, we found that the performance of explicit integration significantly surpasses that of the implicit method. Consequently, we have selected explicit integration as our final approach. Detailed comparison results are presented in Sec. 4.3.

After the aggregation and extraction of object-level features, $\mathbf{E}_{\mathrm{F}}$ and $\mathbf{E}_{\mathrm{B}}^*$ are processed individually by two separate projectors ($p_F$ and $p_B$) to align with the text embeddings $\mathbf{E}_{\mathrm{T}}$. The aligned features are then concatenated as input for LLM. We try multiple strategies to merge object-level features into visual embeddings and find the concatenation operation yields the most beneficial results. The experiments are discussed in Sec. 4.3. During training, we optimize Eq. (6) on the visual instruction tuning data to enhance the multi-modal comprehension capabilities of MG-LLaVA. We execute the aforementioned operations for video training to each frame and then concatenate the results into an extended sequence.

$$p\left(\mathbf{X}_{\mathrm{A}} \mid V_L, V_H, B, T\right) = \prod_{i=1}^{L} p_\theta \left(\mathbf{X}_{\mathrm{A}}^{[i]} \mid Concat(p_F(\mathbf{E}_{\mathrm{F}}), p_B(\mathbf{E}_{\mathrm{B}}^*), \mathbf{E}_{\mathrm{T}}^{[1:i-1]}), \mathbf{X}_{\mathrm{A}}^{[i-1]}\right) \tag{6}$$

### 3.3 MODEL TRAINING AND INFERENCE

Recently, a variety of powerful tagging models and open-vocabulary detectors have emerged, demonstrating remarkable efficacy. By using one specific tagging model to output labels, which are then used by the detector to generate bounding boxes, we can effectively avoid the generation of numerous irrelevant boxes, contrasting with the direct use of class-agnostic detectors. The details of the inference pipeline are illustrated in Appx. D. For the acquisition of object bounding boxes, we employ the well-pretrained RAM (Zhang et al., 2023e) as the tagging model and OWL-ViT v2 (Minderer et al., 2024) as the detector. The generated bounding boxes are filtered by NMS and then fed to models for training and inference. It is important to note that while the RAM model aids in generating tags, these tags serve solely as inputs for the open-vocabulary detector to determine the bounding boxes and are not integrated into the training phase. For video inference, we detect bounding boxes for each frame and concatenate the object queries with the corresponding frame's visual sequence.

Following LLaVA-1.5 (Liu et al., 2023a), we conduct a two-stage training process. During the pretraining stage, we freeze all visual encoders and the LLM and only train the fusion module, visual projector, and box projector. This aims to refine the fusion module's capability to aggregate features of low and high resolutions and to enhance the projector's alignment of visual features with the text embeddings. During instruction tuning, we freeze the visual encoders to maintain the integrity of high-quality image feature extraction and fine-tune the remaining components to enhance multi-modality comprehension.

## 4 EXPERIMENTS

### 4.1 IMPLEMENTATION DETAILS

**DetailedModel Settings.** In this work, all experiments are conducted based on Xtuner (Contributors, 2023). Specially, we choose CLIP pre-trained ViT-Large-14-336 (Radford et al., 2021) as a low-resolution visual encoder and the LAION pre-trained ConvNext-Large-320 (Schuhmann et al., 2022) for high-resolution vision encoder. For the generation of bounding boxes, we have selected RAM-Plus (Zhang et al., 2023e) as the tagging model and OWL-ViTv2-large-patch14-ensemble (Minderer et al., 2024) as the open-vocabulary detector.

**Datasets.** During the image-based training stage, our dataset comprises 558K image-caption pairs from LAION-CCSBU (Sharma et al., 2018) and 708k image-caption pairs from ALLaVA-4V-Caption

Table 1: Comparison with leading methods on several popular visual benchmarks that concentrate on perception. **Params.** denotes the total number of parameters within the model. **Res.** refers to the resolution of the input image, which is assumed to be square by default unless otherwise indicated. The notation '()' signifies the presence of both low-resolution and high-resolution inputs, with the number inside the parentheses specifying the higher resolution.

| Method | LLM | Param. | Data | Res. | MMB$^D$ | MMB$^T$ | SEED$^I$ | MMStar |
|---|---|---|---|---|---|---|---|---|
| *Private Models* | | | | | | | | |
| GPT-4V (OpenAI, 2023) | - | - | - | - | 75.1 | 77.0 | 72.3 | 49.7 |
| GeminiProVision (Team et al., 2023) | - | - | - | - | 75.2 | 73.6 | 70.7 | 38.6 |
| Qwen-VL-Plus (Bai et al., 2023) | - | - | - | - | 66.2 | 67.0 | 65.7 | 39.7 |
| *Open-source Models* | | | | | | | | |
| BLIP-2 (Li et al., 2023b) | Vicuna-13B | 14.2B | 129M | 224 | - | - | 46.4 | - |
| InstructBLIP (Dai et al., 2024) | Vicuna-7B | 8.2B | 130M | 224 | - | 36 | 53.4 | - |
| Shikra (Chen et al., 2023a) | Vicuna-13B | 7.3B | 6M | 224 | 58.8 | 60.2 | - | - |
| IDEFICS-80B (Laurençon et al., 2024) | LLaMA-65B | - | - | 224 | - | 54.6 | - | - |
| Qwen-VL (Bai et al., 2023) | Qwen-7B | 9.6B | 1.4B | 448 | 38.2 | 32.2 | 56.3 | - |
| Qwen-VL-Chat (Bai et al., 2023) | Qwen-7B | 9.6B | - | 448 | 60.6 | 61.8 | 58.2 | 37.5 |
| LLaVA-1.5 (Liu et al., 2023a) | Vicuna-7B | 7.2B | 1.2M | 336 | 65.2 | 66.5 | 66.1 | 30.3 |
| LLaVA-1.5 (Liu et al., 2023a) | Vicuna-13B | 13.4B | 1.2M | 336 | 69.2 | 69.2 | 68.2 | 32.8 |
| LLaVA-HR (Luo et al., 2024) | Vicuna-7B | 7.4B | 1.2M | 448 (1024) | - | - | 64.5 | - |
| SPHINX (Lin et al., 2023b) | Vicuna-7B | 10B | 1.0B | 224 | 66.9 | - | 69.1 | - |
| SPHINX-1k (Lin et al., 2023b) | Vicuna-7B | 10B | 1.0B | 448 | 67.1 | - | 71.6 | - |
| MiniCPM-V2 (Hu et al., 2024) | MiniCPM-2.4B | 2.8B | - | 448 | 69.6 | 69.1 | 67.1 | 39.1 |
| MOVA (Zong et al., 2024) | Vicuna-7B | 10B | 16.6M | 576 | 70.4 | - | - | - |
| LLaVA-UHD Xu et al. (2024) | Vicuna-13B | 13.4B | 1.2M | 672×1008 | 68.0 | - | - | - |
| LLaVA-HR (Luo et al., 2024) | Vicuna-7B | 7.4B | 1.2M | 1024 | - | - | 64.2 | - |
| Mini-Gemini (Li et al., 2024c) | Vicuna-7B | 7.4B | 2.7M | 336 (768) | 69.3 | 68.2 | 68.9 | 37.6 |
| *Our Models* | | | | | | | | |
| MG-LLaVA | Phi3-3.8B | 4.2B | 2.5M | 336 (768) | 74.2 | 74.4 | 70.3 | 41.3 |
| MG-LLaVA | Vicuna-7B | 7.4B | 2.5M | 336 (768) | 72.1 | 71.9 | 69.4 | 35.1 |
| MG-LLaVA | LLaMA3-8B | 8.4B | 2.5M | 336 (768) | 76.5 | 76.6 | 71.5 | 36.9 |
| MG-LLaVA | Vicuna-13B | 13.6B | 2.5M | 336 (768) | 72.2 | 73.5 | 70.8 | 34.1 |
| MG-LLaVA | Yi1.5-34B | 34.4B | 2.5M | 336 (768) | **80.1** | **79.1** | **73.7** | **47.9** |

dataset (Chen et al., 2024a), culminating in a total of 1.2M image-caption pairs for pretraining. The datasets employed for instruction-tuning encompass 665K mixture dataset from LLaVA-Instruct (Liu et al., 2023a), 692k instructions from ALLaVA-4V-Instruction dataset (Chen et al., 2024a), and an additional 25k instructions derived from a combination of ShareGPT4V (Chen et al., 2023b), DocVQA (Tito et al., 2021), DVQA (Kafle et al., 2018) and AI2D (Kembhavi et al., 2016), with a total number of more than 1.3M image-text conversations. The superior quality of this dataset contributes to a swift enhancement in performance. For video training, following Video-LLaVA (Lin et al., 2023a), we combine 558K image-text pairs and 703k video-text pairs for video pertaining. For instruction-finetuning, we utilize a 665k image-text instruction dataset from LLaVA and a 100k video-text instruction dataset from Video-ChatGPT (Maaz et al., 2023).

**Training Details.** We fix all seeds across the training procedures for fairness, where we adopt the XTuner codebase (Contributors, 2023). We established the low-resolution parameter at 336 and the high-resolution parameter at 768. For video training, we uniformly extract 8 frames from each video. During the pretraining stage, we employ a batch size of 32 per device and an aggregate batch size of 256. In the instruction-tuning phase, we reduce the batch size to 16 per device and an overall batch size of 128. The initial learning rate is set to 1e-3 for the pretraining stage and 2e-5 for the instruction-tuning stage. The number of bounding boxes per image is limited to 100 during training. The entire training process takes approximately 23 hours when using the Vicuna7B (Chiang et al., 2023) model using 8×A100 GPUs. For our most extensive model, the Yi1.5-34B (Young et al., 2024), we utilize 32×A100 GPUs and finalize the optimization process in roughly three days by employing the DeepSpeed Zero3 strategy.

## 4.2 MAIN RESULTS

**Perception Benchmarks.** In Tab. 1, we compare our MG-LLaVA with previous leading approaches across several settings on Multi-Modal benchmarks, which mainly concentrate on perception capability, including MMBench-Dev and MMBench-Test (Liu et al., 2023c), SEEDBench-Image (Li et al., 2023a), and MMStar (Chen et al., 2024b). MMBench is dedicated to advancing the understanding

Table 2: Comparison with leading methods on popular VQA visual benchmarks.

| Method | LLM | Param. | Res. | VQA$^T$ | DocVQA | SQA$^I$ | AI2D | VQAv2 | MMVet | LLaVA-w | MMVP |
|---|---|---|---|---|---|---|---|---|---|---|---|
| *Private Models* | | | | | | | | | | | |
| GPT-4V | - | - | - | 78.0 | 42.3 | 82.1 | - | - | 67.7 | - | 38.7 |
| GeminiProVision | - | - | - | 74.6 | - | 81.4 | - | - | 64.3 | - | 40.7 |
| Qwen-VL-Plus | - | - | - | 78.9 | 82.2 | 73.4 | - | - | 61.1 | - | - |
| *Open-source Models* | | | | | | | | | | | |
| BLIP-2 | Vicuna-13B | 14.2B | 224 | 42.5 | - | 61.0 | - | 41.0 | 22.4 | 38.1 | - |
| InstructBLIP | Vicuna-7B | 8.2B | 224 | 50.1 | 10.9 | 60.5 | 40.6 | - | 26.2 | 60.9 | - |
| Shikra | Vicuna-13B | 7.3B | 224 | - | - | - | - | - | - | - | - |
| IDEFICS-80B | LLaMA-65B | - | 224 | 30.9 | - | - | 54.8 | 60 | - | - | - |
| Qwen-VL | Qwen-7B | 9.6B | 448 | 63.8 | **62.1** | 67.1 | 57.7 | 78.8 | - | - | - |
| Qwen-VL-Chat | Qwen-7B | 9.6B | 448 | 61.5 | 57.1 | 68.2 | 63 | 78.2 | - | - | - |
| LLaVA-1.5 | Vicuna-7B | 7.2B | 336 | 58.2 | 21.5 | 66.8 | 55.5 | 78.5 | 31.1 | 65.4 | 27.4 |
| LLaVA-1.5 | Vicuna-13B | 13.4B | 336 | 61.3 | 24.1 | 71.6 | 61.1 | 80.0 | 36.1 | 72.5 | - |
| SPHINX | Vicuna-7B | 10B | 224 | 51.6 | - | 69.3 | - | 78.1 | 36.0 | 73.5 | - |
| SPHINX-1k | Vicuna-7B | 10B | 448 | 58.8 | - | 69.1 | - | 80.2 | 36.6 | 74.3 | - |
| LLaVA-UHD | Vicuna-13B | 13.4B | 672×1008 | 67.7 | - | 72.0 | - | 81.7 | - | - | - |
| LLaVA-HR | Vicuna-7B | 7.4B | 1024 | 67.1 | - | 65.1 | - | 81.9 | 31.2 | - | - |
| Mini-Gemini | Vicuna-7B | 7.4B | 336(768) | 65.2 | - | - | - | - | 40.8 | - | 35.3 |
| *Our Models* | | | | | | | | | | | |
| MG-LLaVA | Phi3-3.8B | 4.2B | 336(768) | 66.4 | 49.1 | 74.5 | 74 | 80.1 | 47.3 | 75.4 | **50.0** |
| MG-LLaVA | Vicuna-7B | 7.4B | 336(768) | 67.3 | 47.9 | 70.8 | 69.3 | 80.2 | 41.0 | 75.5 | 47.3 |
| MG-LLaVA | LLaMA3-8B | 8.2B | 336(768) | 68.1 | 49.0 | 76.3 | 75.6 | 80.7 | 46.9 | 75.5 | 37.3 |
| MG-LLaVA | Vicuna-13B | 13.6B | 336(768) | 69.6 | 52.1 | 74.7 | 73.4 | 81.2 | 46.7 | **82.0** | 40.7 |
| MG-LLaVA | Yi1.5-34B | 34.4B | 336(768) | **70.0** | **56.1** | **77.0** | **81.1** | **82.0** | **48.4** | 80.5 | **50.0** |

of multi-modal perception and cognition, and SEEDBench provides a comprehensive and objective evaluation of MLLM. MMStar further ensures each selected sample exhibits visual dependency. MG-LLaVA exhibits a significantly enhanced perceptual capability compared to a wide range of MLLMs. Our MG-LLaVA equipped with phi3-3.8B (Abdin et al., 2024) show superior performance than MiniCPM V2 (Hu et al., 2024) of +4.6%/5.3% on MMBench Dev/Test, and +3.2% on SEEDBench. Utilizing Vicuna-7B (Chiang et al., 2023), MG-LLaVA outperforms all models with vicuna-7B and even 13B on MMBench and SEEDBench, surpassing LLaVA-1.5-7B by an average of 5.1% across four benchmarks. Moreover, with Yi1.5-34B (Young et al., 2024), MG-LLaVA consistently outperforms GPT-4V on MMBench and SEEDBench. Concurrently, it maintains equivalent efficacy to GPT-4V on MMStar. Incorporating multi-granularity visual inputs, MG-LLaVA develops its capability of capturing details within the image. More cases are exhibited in Appx. B.

**Visual Question Answering Benchmarks.** In this section, we analyze MLLM's capability of visual conversation. The benchmarks can be divided into two groups: (1)Benchmarks require understanding the text within images to provide answers, including TextVQA(VQA$^T$) (Singh et al., 2019) and DocVQA (Mathew et al., 2021). We report the accuracy of both validation sets. (2)General visual question answering benchmarks such as VQA-V2 (Antol et al., 2015), ScienceQA-Image(SQA$^I$) (Lu et al., 2022), AI2D (Kembhavi et al., 2016), MMVet (Yu et al., 2023), LLaVA-W (Liu et al., 2023a), and MMVP(Tong et al., 2024). The evaluation results on VQA benchmarks are shown in Tab. 2. MG-LLaVA also demonstrates considerable proficiency on VQA benchmarks. When equipped with Vicuna-7B and 7.4B parameters, MG-LLaVA surpasses both SPHINX-1k (Lin et al., 2023b), which has 10B parameters, and Mini-Gemini with 7.4B parameters on these benchmarks, despite utilizing even less data. Operating under identical parameter conditions, MG-LLaVA-Vicuna13B, with low-resolution input of 336 and high-resolution of 768, outperforms LLaVA-UHD (Xu et al., 2024), which incorporates an input resolution of 672×1008 on VQA$^T$, SQA$^I$, and AI2D. Additionally, MG-LLaVA demonstrates significant improvement on the MMVP benchmark, which is particularly challenging for MLLMs. MG-LLaVA-Vicuna-7B achieves an accuracy of 47.3, surpassing Mini-Gemini's score of +12% and even exceeding that of GPT-4V. MG-LLaVA exhibits its potential for expansion when integrated with larger LLM. With Yi1.5-34B (Young et al., 2024), MG-LLaVA surpasses the majority of established baselines across a wide array of VQA benchmarks.

**Video Question Answering Benchmarks.** To demonstrate the effectiveness of our approach, we have expanded our model to encompass video comprehension. We evaluate our models on MSVD and MSRVTT, and results are shown in Tab. 3. MG-LLaVA outperforms Video-LLaVA (Lin et al., 2023a) on both benchmarks, which further proves the efficiency of MG-LLaVA. In video understanding, MG-LLaVA demonstrates proficiency in identifying the critical object in the video. More illustrative instances are depicted in Appx. B.

Table 3: Comparison with other methods on Video-QA benchmarks.

| Method | LLM | MSVD-QA | MSRVTT-QA |
|---|---|---|---|
| FrozenBiLM (Yang et al., 2022) | - | 32.2 | 16.8 |
| VideoChat (Li et al., 2023c) | Vicuna-7B | 56.3 | 45.0 |
| LLaMA-Adapter (Zhang et al., 2023d) | - | 54.9 | 43.8 |
| Video-LLaMA (Zhang et al., 2023a) | Vicuna-7B | 51.6 | 29.6 |
| Video-ChatGPT (Maaz et al., 2023) | Vicuna-7B | 64.9 | 49.3 |
| Video-LLaVA (Lin et al., 2023a) | Vicuna-7B | 70.7 | 59.2 |
| MG-LLaVA | Vicuna-7B | **71.5** | **59.8** |

Table 4: Ablation results on MMBench-DEV, TextVQA, and GQA. **Params.** denotes the number of model parameters, while **Inf. Speed** represents the speed of inference. We execute our baseline based on the LLaVA model on the Xtuner codebase with Vicuna-7B and Phi3-3.8B.

| Object-level Features | Conv-Gate Fusion | Vicuna-7B | | | | | | Phi3-3.8B | | | | | |
|---|---|---|---|---|---|---|---|---|---|---|---|---|---|
| | | #TFLOPS | Params. | Inf. Speed | MMB$^D$ | VQA$^T$ | GQA | #TFLOPS | Params. | Inf. Speed | MMB$^D$ | VQA$^T$ | GQA |
| × | × | 5.76 | 7.2B | 8.89 tokens/s | 69.5 | 60.5 | 59.3 | 3.3 | 4.0B | 35.00 tokens/s | 70.7 | 58.1 | 58.3 |
| ✓ | × | 6.20 | 7.4B | 8.71 tokens/s | 70.6(+1.1) | 61.0(+0.5) | 60.3(+1.0) | 3.72 | 4.2B | 34.54 tokens/s | 73.0(+2.3) | 59.0(+0.9) | 59.1(+0.8) |
| ✓ | ✓ | 6.21 | 7.4B | 8.46 tokens/s | 72.1(+2.6) | 67.3(+7.8) | 61.3(+2.0) | 3.73 | 4.2B | 34.04 tokens/s | 74.2(+3.5) | 66.4(+8.3) | 60.4(+2.1) |

Table 5: Comparison with different MLLM designs.

| Method | LLM | MMB$^D$ | MMStar | VQA$^T$ | GQA |
|---|---|---|---|---|---|
| LLaVA-HR | Phi3-3.8B | 72.2 | 38.4 | 65.9 | 59.7 |
| Mini-Gemini | Phi3-3.8B | 73.2 | 39.5 | **66.4** | 59.7 |
| MG-LLaVA | Phi3-3.8B | **74.2** | **41.3** | **66.4** | **60.4** |

Table 6: Results of explicit and implicit integration of object-level features.

| Method | LLM | MMB$^D$ | MMStar | VQA$^T$ | GQA |
|---|---|---|---|---|---|
| Implicit Integration | Vicuna-7B | 70.8 | 34.7 | 66.8 | 61.3 |
| Explicit Integration | Vicuna-7B | 72.1 | 35.1 | 67.3 | 61.3 |

## 4.3 ABLATION EXPERIMENTS

In this section, we conduct comprehensive ablation studies of our model. The ablation experiments are based on Xtuner codebase (Contributors, 2023), with a fixed seed protocol to ensure the stability and comparability of the experimental conditions.

**Effect of Each Component.** We first conduct ablation studies on object-level features and the Conv-Gate fusion module across multiple datasets of different purpose, including MMBench-DEV (Liu et al., 2023c), TextVQA (Singh et al., 2019), and GQA (Hudson & Manning, 2019). To validate the effectiveness of our method on different scales of LLM, the baseline is built on Vicuna-7B and Phi3-3.8B using the Xtuner codebase. The training data and seed are consistently set to ensure fairness. The results are shown in Tab. 4.

It is clear that the model achieves significant gains with the integration of object-level features and the Conv-Gate Fusion module. When adding object-level features, the performance of MMBench-Dev, GQA increases 1.1%, 1.0% separately with Vicuna-7B and 2.3%, 0.8% with Phi3. After utilizing the fusion network, the performance on these two benchmarks further increases by 2.6%, 2.0% with Vicuna-7B and 3.5%, 2.1% with Phi3. For the TextVQA benchmark, the incorporation of object-level features does not markedly enhance performance due to the suboptimal detection of textual content within images by the detector. Nevertheless, the integration of high-resolution features mitigates this limitation, culminating in an accuracy increment of 7.8% on Vicuna-7B and 8.3% on Phi3-3.8B. The integration of both modules incurs a marginal increase in computational expense and parameter count, yet it enhances the efficacy of models across various scales. We further enumerate additional comparative outcomes across various subsets of MMBench-Dev, the comparative results are shown in Appx. A.

**Comparison with Other MLLM Design.** To demonstrate the efficiency of our framework, we reconstruct two fusion-based MLLM, Mini-Gemini (Li et al., 2024c) and LLaVA-HR (Luo et al., 2024) on Xtuner codebase and conduct a comparative analysis of these two multi-input methods against MG-LLaVA. We conduct the experiments on Phi3-3.8B. Specifically, we integrate the fusion module of LLaVA-HR into the 12th layer of the visual encoder. To ensure a fair comparison, the input resolutions are standardized. The results, detailed in Tab. 5, indicate that our multi-granularity vision flow outperforms complex fusion-based models across multiple downstream tasks.

Table 7: Comparison of different fusion modules, methods of merging object-level features, and tagging models.

(a) Fusion modules.

| Method | MMB$^D$ | MMStar |
|---|---|---|
| Baseline | 69.2 | 34.1 |
| w/ *Resampler* | 55.6 | 30.5 |
| w/ *Channel Concat* | 68.9 | 32.6 |
| w/ *Patch Info Mining* | 68.3 | 32.9 |
| w/ *Conv-Gate Fusion* | **69.8** | **34.5** |

(b) Methods of merging object-level features.

| Method | MMB$^D$ | MMStar |
|---|---|---|
| Baseline | 68.2 | 32.5 |
| w/ *F-to-B Cross Attention* | 65.7 | 33.3 |
| w/ *B-to-F Cross Attention* | 67.7 | 34.4 |
| w/ *Concat* | **69.8** | **34.5** |

(c) Tagging models.

| Method | MMB$^D$ | MMStar |
|---|---|---|
| Baseline | 68.2 | 32.5 |
| w/ *COCO80* | 68.3 | 32.9 |
| w/ *RAM tags* | **69.2** | **34.5** |

**Fusion Network Design.** We also explore a diverse design of fusion modules and perform ablation studies on various components: (1)*Channel Concat.* We simply concat the low and high-resolution features in the channel dimension. (2) *Patch Info Mining*. We replace the gated-fusion model with Patch Info Mining in (Li et al., 2024c). (3) *Resampler.* We substitute the gated-fusion model with a resampler in (Alayrac et al., 2022). The results are shown in Tab. 7a. We find our Conv-Gated fusion module performs better through these methods, which confirms its efficiency.

**Method of Merging Object-level Features.**

(1) We first compare the performance of explicit integration and implicit integration. The results are presented in Tab. 6. It can be observed from the table that the explicit method demonstrates superior performance compared to the implicit method across various benchmarks.

(2) Based on the explicit integration method, we further explore various methods for incorporating object-level features: (1)*F-to-B Cross-Attention*. We add a cross-attention block to enhance the fusion features by integrating object-level features after the fusion module, the enhanced fusion features are then fed into LLM. (2)*B-to-F Cross-Attention*. Following the fusion module, another cross-attention block is employed to enhance the object-level features by integrating fusion features. The fusion features and enhanced object-level features are then concatenated as input for LLM. The frameworks of both are depicted in Appx. C, and the results are reported in Tab. 7b. Our observations indicate that cross-attention does not enhance the integration of object-level features into visual representations. Conversely, concatenating object-level features with visual tokens and deferring the decision-making to the LLM yields more favorable outcomes.

**Tagging Model.** We investigate the impact of the tagging model within the bounding box generation pipeline. We compare our method with assigning fixed tags based on the 80 categories from the COCO (Lin et al., 2014) dataset to open-vocabulary detectors for producing bounding boxes. The comparative results are presented in Tab. 7c. Given that the COCO dataset's 80 categories do not comprehensively cover real-world objects, the generated bounding boxes fail to encompass all objects within an image. This limitation consequently diminishes the impact of object-level features.

## 5 DISCUSSIONS

**Conclusions.** In this work, we propose MG-LLaVA, an expansive multi-modal model adept at processing visual inputs of multiple granularities, encompassing object-level features, original images, and high-resolution data. To effectively amalgamate features of varying granularities, we propose the Multi-Granularity Vision Flow module, thereby equipping the LLM with the ability to discern multi-modal interactions from a consolidated visual framework. Utilizing a range of LLMs extending from 3.8B to 34B parameters, our model exhibits pronounced scalability and remarkable performance in visual understanding, outperforming established models and significantly outperforming GPT-4V and GeminiPro Vision on benchmarks such as MMBench and SEEDBench. The validity of our methodology is substantiated through rigorous empirical studies. MG-LLaVA establishes a foundational baseline for future explorations into more sophisticated techniques of integrating inputs of multiple granularities.

**Broader Impacts.** As a robust multi-modal language model, MG-LLaVA exhibits considerable prowess in visual perception and comprehension, offering an innovative methodology to refine MLLMs further. However, MG-LLaVA's potential societal implications merit attention, as it may facilitate the creation of multimodal applications, including those with possible adverse effects.

**Reproducibility Statement**

We have included all of our code in the supplementary materials, encompassing training, evaluation, and inference. Additionally, we provide our training script and seed to ensure the reproducibility of our method.

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

## A  APPENDIX / DETAILED RESULTS ON SUBSETS

In this section, we compare the influence of object-level features on several subsets of MMBench-Dev and Seed-bench, as shown in Fig. 5. It can be observed that the integration of object-level features significantly enhances the model's capability in multiple aspects of perception including Attribute Reasoning, Fine-grained Perception, Physical Relation Perception, Visual Reasoning, *etc*.

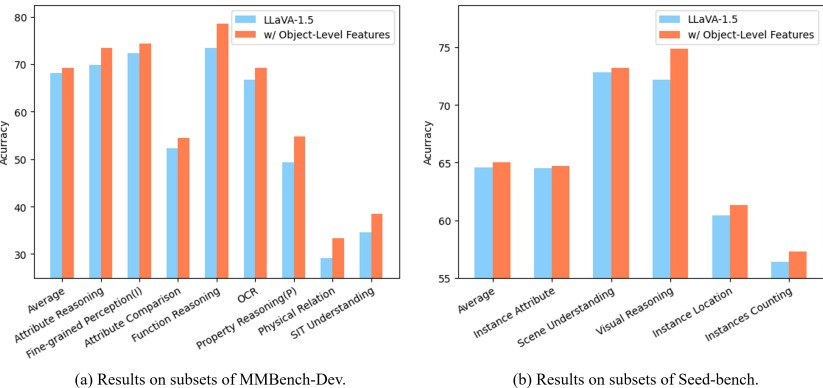

(a) Results on subsets of MMBench-Dev.                    (b) Results on subsets of Seed-bench.

Figure 5: Ablation study on several subsets of MMBench-DEV-EN and Seed-bench. Fine-grained Perception(I) denotes *Fine-grained Perception(instance-level)*, Property Reasoning(P) means *Property Reasoning Perception* and SIT Understanding denotes *Structuralized Image-Text Understanding*.

## B  APPENDIX / ADDITIONAL SHOWCASES

In this section, we present additional instances to substantiate the capability of MG-LLaVA. As presented in Fig. 6 and Fig. 7, MG-LLaVA is proficient in addressing queries that necessitate meticulous attention to specifics and in capturing fine-grained details within image or video. These further instances reinforce the superior performance of our MG-LLaVA in visual comprehension.

## C  APPENDIX / METHOD OF MERGING OBJECT-LEVEL FEATURES

The illustration of *F-to-B Cross-Attention* and *B-to-F Cross-Attention* is depicted in Fig. 8.

## D  APPENDIX / INFERENCE PIPELINE

The inference pipeline of MG-LLaVA is displayed in Fig. 9. The tagging model first processes the input image to provide tags within the image, which are subsequently utilized as the text input of the detector to derive bounding boxes corresponding to the tagged obejcts within the image.

## E  APPENDIX / COMPARISON OF TAGGING MODELS

Table 8: Ablation results on MMBench-DEV Liu et al. (2023c), SEEDBench Li et al. (2023a) and TextVQA Singh et al. (2019). We execute our experiments based on the LLaVA model with Vicuna-7B and Phi3-3.8B.

| Method | Images | Number of Boxes | | | | | |
|---|---|---|---|---|---|---|---|
| | | 0 | 1-10 | 12-20 | 21-30 | 30-50 | ¿50 |
| COCO 80 + OWL-ViT v2 | 389722 | 71118 | 245952 | 44059 | 28593 | 0 | 0 |
| RAM + OWL-ViT v2 | 389722 | 43654 | 184706 | 91245 | 34648 | 22827 | 12645 |

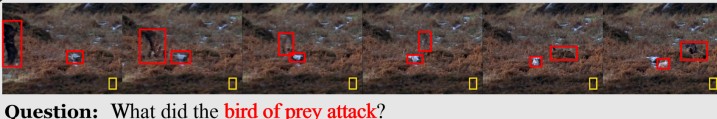

**Question:** What did the **bird of prey attack**?

Video-LLaVA:
The bird of prey attacked **a small bird** in the video.

🦙MG-LLaVA:
The bird of prey attacked a rabbit, which was hiding in the grass. The rabbit was startled and ran away.

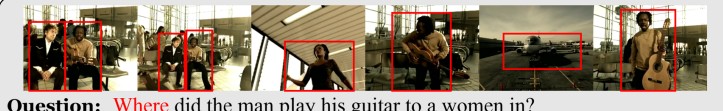

**Question:** Where did the man play his guitar to a women in?

Video-LLaVA:
The man played his guitar to a woman **in a plane**, as seen in the video.

🦙MG-LLaVA:
The man played his guitar to a woman in the airport. He played a song for her and sang along.

Figure 6: More cases of video understanding.

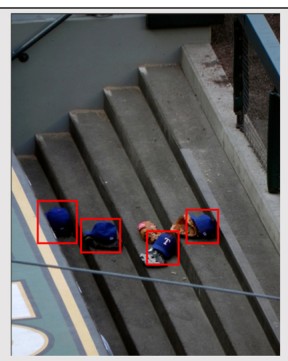

**Question:** How many **baseball caps** are there?

LLaVA:
There are **several baseball caps** scattered on the bleachers, with some of them being blue and white.

🦙MG-LLaVA:
There are **four baseball caps** visible in the image. Three of them are blue, and one is red with a white \"T\" logo, which is likely the logo of the Texas Rangers, a Major League Baseball team. The caps are lying on the steps, and their positioning suggests they may have been discarded or forgotten by their owners. The caps are in various states of wear, with the red cap showing more signs of wear and tear, possibly from frequent use or exposure.

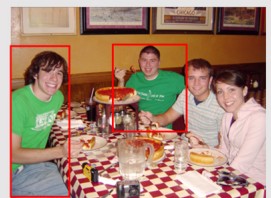

**Question:** How many people are wearing green t-shirts?

LLaVA:
There are **three people** wearing green t-shirts in the image. They are sitting around a dining table, enjoying a meal together.

🦙MG-LLaVA:
In the image, there are **two people** wearing green t-shirts. The first individual is a young man on the left side of the image, who is smiling and appears to be in mid-conversation or laughter. He is wearing a green t-shirt with a graphic on the front, which is not fully visible in the image. The second individual is a young man in the center of the image, who is also smiling.

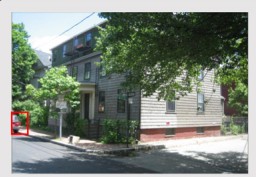

**Question:** Is there **a red car** in the image?

LLaVA:
There is no red car in this image.

🦙MG-LLaVA:
Yes, there is a red car in this image.

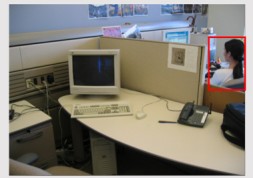

**Question:** Is there **a woman** in the image?

LLaVA:
There is no woman in this image.

🦙MG-LLaVA:
Yes, there is a woman in this image.

Figure 7: More cases of image understanding.

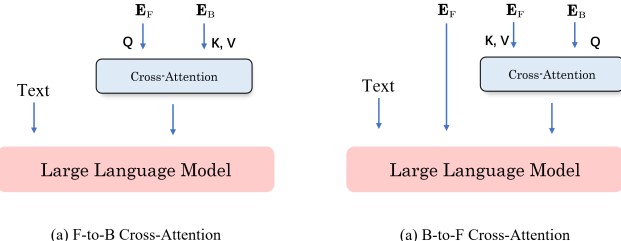

(a) F-to-B Cross-Attention    (a) B-to-F Cross-Attention

Figure 8: Illustration of F-to-B Cross-Attention and B-to-F Cross-Attention.

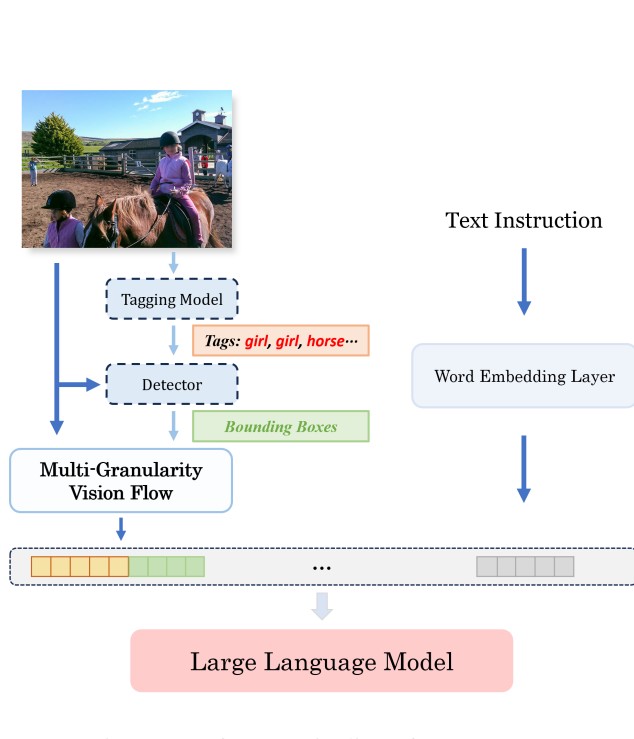

Figure 9: Inference pipeline of MG-LLaVA.

