# OpenReview forum: "MG-LLaVA: Towards Multi-Granularity Visual Instruction Tuning"
_ICLR.cc/2025/Conference — ICLR 2025 Conference Withdrawn Submission_

### Official Review · Reviewer_8mda · 2024-10-28

**Soundness:** 2
**Presentation:** 2
**Contribution:** 2
**Rating:** 5
**Confidence:** 5

**Summary:**

Summary:

MG-LLaVA is a multi-modal large language model (MLLM) designed to improve visual processing capabilities by using a multi-granularity vision flow. This includes low-resolution, high-resolution, and object-centric features to enhance perception tasks requiring detailed visual information. Extensive experiments have validated its effectiveness.

Contributions:

1. Leveraging an additional open vocabulary detection model introduces multi-granularity object-level features to enhance the perceptual capabilities of MLLMs.
2. Extensive experiments demonstrate the effectiveness of the method.

**Strengths:**

1. The structure of paper is simple and easy to read, and the model implementation is very easy to follow.
2. The idea is very straightforward, and the experiments are solid. It is reasonable to introduce multi-granularity object-level features to enhance the perceptual capabilities of Multimodal Large Language Models (MLLMs).

**Weaknesses:**

1. The idea appears incremental, as it simply integrates high-resolution image interpretation with region-level image understanding, resembling a trick
2. Experimental evaluations and fair comparisons are notably lacking. Given that multi-granularity features are utilized to augment the model's perceptual abilities, evaluations should be conducted on fine-grained perception datasets. General VQA is inadequate for assessing the fine-grained perceptual capabilities of MLLM.
3. Excessive reliance on additional detector inputs may result in suboptimal, non-end-to-end outcomes.

**Questions:**

1. Although the method performs well on the general VQA, it lacks a comprehensive assessment of fine-grained perception capabilities. It would be more fair and convincing to compare it with region-level methods like Next-Chat and Osprey on the RefCOCO dataset. This could be accomplished by using the bounding box of the corresponding target as input.

2. It is evident that using object-level features can enhance the perception ability of MLLMs. However, incorporating additional detectors introduces extra computational costs and biases. An equitable efficiency comparison is necessary. if these added costs surpass the benefits from data expansion, parameter extension, or data filtering, it results in negative optimization, as I believe is the case with MG-LLaVA. From the performance comparison, when using Vicuna 7B, MMStar exhibits lower performance than other models, indicating data leakage risk and validating the risk of bias introduced by reliance on detectors.

3. Although MG-LLaVA shows improvements in general capabilities, these enhancements are marginal. The added expense of using additional detection models and object-level features should yield a greater performance boost. Moreover, during inference, reliance on detection results from other models is cumbersome. Transforming external dependencies into self-mining processes could significantly enhance the practical utility of the model.

---

### Official Review · Reviewer_uMyd · 2024-10-30

**Soundness:** 3
**Presentation:** 4
**Contribution:** 2
**Rating:** 5
**Confidence:** 3

**Summary:**

This paper aims to improve the visual capabilities of MLMs (multimodal large models) by proposing a new model MG-LLaVA. limited to resources, most of MLMs nowadays just have low resolution inputs, which are challenging on fine-grained tasks. Therefore, this papet proposes a novel framework that introduces object-level feature in addition to high resolution visual encoder. Based these, the article also uses a gating-based fusion strategy as well as explicit integration on object-level feature. These approaches reduce the computational pressure introduced by high resolution images and simultaneously improve performance on fine-grained tasks. On MMBench and SEEDBench, the model outperforms even the private models GPT-4V and GeminiPro-V. The article also conducts extensive experiments to show that their framework achieves competitive scores on multiple datasets of images or videos.

**Strengths:**

The goal of this paper is to release the power of MLMs on fine-grained tasks. A high resolution visual encoder is introduced to make up for the complement of previous work. And some fusion and compression strategies are introduced to ease the computational pressure. In addition to this, the article demonstrates that this new framework achieves significantly higher scores on MLMs at several scales, which fully demonstrates the effectiveness of the method. Moreover, this is the first approach to introduce object-level features in the field of MLMs, and experimentally, the article demonstrates the ability of their method to achieve higher scores than private models under MMBench and SEEDBench.

**Weaknesses:**

1. As mentioned in the article itself, the introduction of multi-granularity and multi-scale to enhance model performance is a common approach to convolutional networks, and merely migrating this approach to the field of MLMs is hardly an innovative contribution. Some of the algorithms used in the article from object detection only do some information enhancement on the input side, while many MLMs can already accomplish the object detection task by themselves nowadays.
2. The scores achieved on both the MMBench as well as SEEDBench datasets, while respectable, are not compared to some of the more competitive models. I identified MMB as version 1 and SEEDBench as Avg based on the scores of Qwen-VL and MiniCPM-V2, and there are a number of scores on both leaderboards that are higher than the scores of MG-LLaVA work, eg. Honeybee (Cha et al., 2024), AllSeeing-v2 (Wang et al. 2024) based on Vicuna-13b at MMB-test. and then you can also find a lot of similar models with higher scores on the same substrate.
3. In addition to Perception Benchmarks. this problem can also be found in Visual QA and Video QA. such as on the MSRVTT-QA dataset. there are already many models with very high scores in 2024. Some of them also use some methods to improve the model's ability on fine-grained tasks. eg. Flash-VStream (Zhang et al. 2024) Monkey (Li et al. 2023). The article does not seem to compare these new 2024 models.

To summarize, I think the approach proposed in the article is valid, but MG-LLaVA does not do the job of making a difference, either from an innovation perspective or from a performance perspective.

[1] Cha, Junbum, et al. "Honeybee: Locality-enhanced projector for multimodal llm." *Proceedings of the IEEE/CVF Conference on Computer Vision and Pattern Recognition*. 2024.

[2] Wang, Weiyun, et al. "The all-seeing project v2: Towards general relation comprehension of the open world." *arXiv preprint arXiv:2402.19474* (2024).

[3] Zhang, Haoji, et al. "Flash-VStream: Memory-Based Real-Time Understanding for Long Video Streams." *arXiv preprint arXiv:2406.08085* (2024).

[4] Li, Zhang, et al. "Monkey: Image resolution and text label are important things for large multi-modal models." *Proceedings of the IEEE/CVF Conference on Computer Vision and Pattern Recognition*. 2024.

**Questions:**

1. The SEEDBench mentioned in the article uses SEEDBench-Image, but I checked the scores for leaderboard and the other methods mentioned in the paper, and they seem to correspond to SEEDBench-Avg (which contains both video and image), so it's not clear to me whether the comparison here includes scores from the video task.
2. If an open vocabulary detector is used, why is a tagger used to determine the bounding box instead of generating ROI directly based on text embedding?
3. The article suggests that this approach is intuitively better for small target comprehension or counting tasks, are there any datasets in this area that show that this approach has more significant performance gains on specific tasks?
4. I found that the Monkey model uses a similar idea to enhance the performance of the model and also proposes to augment the data with traditional CV methods for refinement, is there a comparison to this approach in the paper? For example, changing the base LLM to Qwen-7b to compare with Monkey (Li et al. 2023) and more models on this field.

[1] Li, Zhang, et al. "Monkey: Image resolution and text label are important things for large multi-modal models." *Proceedings of the IEEE/CVF Conference on Computer Vision and Pattern Recognition*. 2024.

---

### Official Review · Reviewer_AQZ3 · 2024-11-01

**Soundness:** 2
**Presentation:** 3
**Contribution:** 2
**Rating:** 3
**Confidence:** 5

**Summary:**

This paper presents a novel Multimodal Large Language Model (MLLM) that improves visual processing capabilities by incorporating a multi-granularity vision flow, which includes low-resolution, high-resolution, and object-centric features. This approach enhances the performance of current Large Language Models (LLMs), as demonstrated in the experiments.

**Strengths:**

1. The integration and fusion of multi-granularity features with object-centric features is novel for MLLMs.
2. Experimental results demonstrate the effectiveness of the proposed pipeline.
3. The paper is well-written and clearly presented.

**Weaknesses:**

1. The performance improvement on similarly sized LLMs in Table 2 and Table 3 appears modest.
2. The ablation study would benefit from visual comparisons to illustrate the impact of each component, such as case studies or visualizations of feature-level effects.
3. Some failure cases should be shown to provide insights into the method’s limitations.
4. It is unclear if the method can handle larger images, such as 1024p or 2k resolutions.

**Questions:**

1. The performance improvement on similarly sized LLMs in Table 2 and Table 3 appears modest.
2. The ablation study would benefit from visual comparisons to illustrate the impact of each component, such as case studies or visualizations of feature-level effects.
3. Some failure cases should be shown to provide insights into the method’s limitations.
4. It is unclear if the method can handle larger images, such as 1024p or 2k resolutions.

---

### Official Review · Reviewer_V5Vw · 2024-11-04

**Soundness:** 2
**Presentation:** 3
**Contribution:** 2
**Rating:** 3
**Confidence:** 5

**Summary:**

The paper proposes MG-LLaVA to improve the capability for recognizing multi-granularity features for current MLLMs and flatten the restriction of the resolution in visual inputs. MG-LLaVA includes the low-resolution, high-resolution, and object-level features altogether and fuses them with a conv-gate fusion module for the general visual features. The object ROIs are extracted by a pre-trained detection model for better object-level understanding skills. The paper proposes a series of MLLMs ranging from 3.8B to 34B based on various LLMs and shows strong performance across image and video multimodal benchmarks.

**Strengths:**

1. The key claim for the paper that multi-granularity features with low-res, high-res, and object features can improve detailed understanding and object recognition skills is reasonable. The authors design the conv-gated fusing module and demonstrate its effectiveness through complete ablation studies.
2. The series of models and benchmarks are clear and complete. The authors train the variants for MG-LLaVA based on Phi, Vicuna, LLaMA3, and Yi1.5 and conduct experiments on various multi-modal benchmarks.
3. The overall architecture of the paper is well-structured and easy to follow.

**Weaknesses:**

1. The idea of fusing multi-granularity features is not novel, as integrating low-resolution and high-resolution images has been demonstrated effect by a range of works, including LLaVA-NeXt, LLaVA-HR, Mini-Gemini, LLaVA-UHD, etc. The difference in MG-LLaVA lies in the usage of detected objects. However, the detection operation introduces extra computational costs and external models with extra information, which is not an optimal solution.
2. The performance comparisons against existing MLLMs are relatively weak. For example, the results of MG-LLaVA equipped with Vicuna-7B do not surpass baselines with similar efforts on some benchmarks, including SQA, TextVQA, MMStar, etc. The overall number of training data is significantly heavier than the baselines (2.6M), which makes the comparisons more unfair.
3. Some key ablations seem lacking. The authors are encouraged to clearly show the contributions for performance with every part of the visual feature to show the difference between previous works.

**Questions:**

The concerning questions are stated in the weakness section. Based on the weaknesses listed above, I lean to reject this manuscript at its current version.

---

### Official Review · Reviewer_hMBr · 2024-11-04

**Soundness:** 3
**Presentation:** 3
**Contribution:** 2
**Rating:** 5
**Confidence:** 5

**Summary:**

This paper presents an MLLM architecture to improve the multi-granularity visual understanding abilities of multimodal models. The method follows the idea proposed in Mini-Gemini to fuse high and low-resolution visual encoders and adds object recognition from other foundation models to enhance object-level understanding ability. A series of models ranging from 3.8B to 34B are proposed and the models are evaluated on multiple popular benchmarks.

**Strengths:**

- The paper is well-organized. The proposed model is evaluated on multiple tasks including general visual understanding benchmarks, VQA, and video datasets. Ablation study and runtime evaluation are also provided.

**Weaknesses:**

- The technical contribution of the paper is not very significant. The paper claims the main contribution is combining low, high-resolution, and object-level features. But the design of combining low and high-resolution features mainly comes from mini-Gemini and some modifications on the fusion module are proposed in the paper. The introduction of object-level features requires extra models and makes the base architecture more complex.

- I am not convinced about the necessity of introducing the extra object-level information. Recent state-of-the-art MLLMs like Qwen2 VL [r1], LLaVA-Onevision [r2] and Pixtral [r3] all adopt a single encoder solution. I think the MLLMs themselves should have the ability to capture object-level information from the images with sufficient data and a proper training strategy. The method proposed in the paper may reduce the requirement for training data but the more complex architecture also makes the solution less general. Besides, the method also didn't show large improvements on SEED (69.4 vs. 68.9) and MMStar (35.1 vs. 37.6) compared to Mini-Gemini with the same base LLM.

- Many recent MLLMs like [r2, r4] are not compared. Compared to these methods, the proposed solution is not strong enough and imore complex.

[r1] Qwen2-VL: Enhancing Vision-Language Model's Perception of the World at Any Resolution

[r2] LLaVA-OneVision: Easy Visual Task Transfer

[r3] Pixtral 12B

[r4] Cambrian-1: A Fully Open, Vision-Centric Exploration of Multimodal LLMs

**Questions:**

Please refer to my comments above.

---

### Note · Authors · 2024-12-14

**Comment:**

We are very grateful for the professional service provided by reviewers throughout the review process and for the valuable feedback offered.

**Withdrawal Confirmation:**

I have read and agree with the venue's withdrawal policy on behalf of myself and my co-authors.